# BUILD-A-SCENE: INTERACTIVE 3D LAYOUT CONTROL FOR DIFFUSION-BASED IMAGE GENERATION

**Abdelrahman Eldesokey & Peter Wonka**
King Abdullah University of Science and Technology (KAUST)
Thuwal, Saudi Arabia
{first.last}@kaust.edu.sa

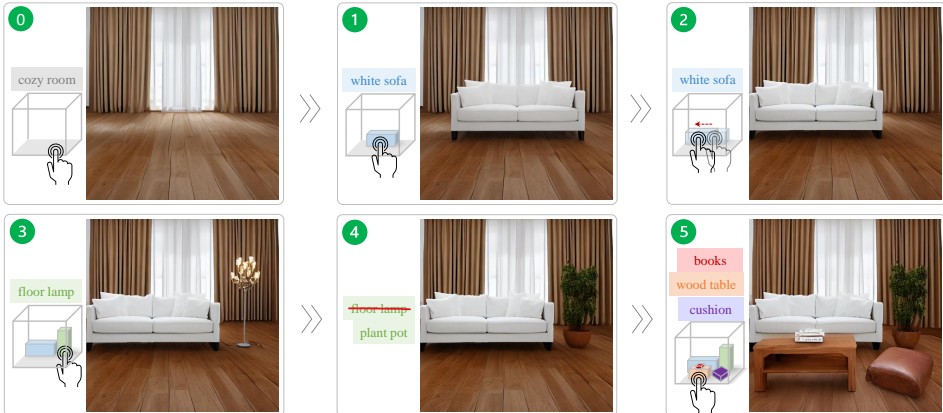

Figure 1: Build-A-Scene is an *interactive* diffusion-based approach for image generation based on a user-provided *3D layout*. At each generation stage, the user can control object type, location, and orientation (in-plane rotation) in 3D. Build-A-Scene ensures that objects are seamlessly integrated into the scene (see shadows and reflections) and preserve their identity under layout changes.

## ABSTRACT

We propose a diffusion-based approach for Text-to-Image (T2I) generation with *interactive 3D layout control*. Layout control has been widely studied to alleviate the shortcomings of T2I diffusion models in understanding objects' placement and relationships from text descriptions. Nevertheless, existing approaches for layout control are limited to 2D layouts, require the user to provide a *static* layout beforehand, and fail to preserve generated images under layout changes. This makes these approaches unsuitable for applications that require 3D object-wise control and iterative refinements, *e.g.*, interior design and complex scene generation. To this end, we leverage the recent advancements in depth-conditioned T2I models and propose a novel approach for interactive 3D layout control. We replace the traditional 2D boxes used in layout control with 3D boxes. Furthermore, we revamp the T2I task as a multi-stage generation process, where at each stage, the user can insert, change, and move an object in 3D while preserving objects from earlier stages. We achieve this through a novel Dynamic Self-Attention (DSA) module and a consistent 3D object translation strategy. To evaluate our approach, we establish a benchmark and an evaluation protocol for interactive 3D layout control. Experiments show that our approach can generate complicated scenes based on 3D layouts, outperforming the standard depth-conditioned T2I methods by *two-folds* on object generation success rate. Moreover, it outperforms all methods in comparison on preserving objects under layout changes. *Project Page:* https://abdo-eldesokey.github.io/build-a-scene/

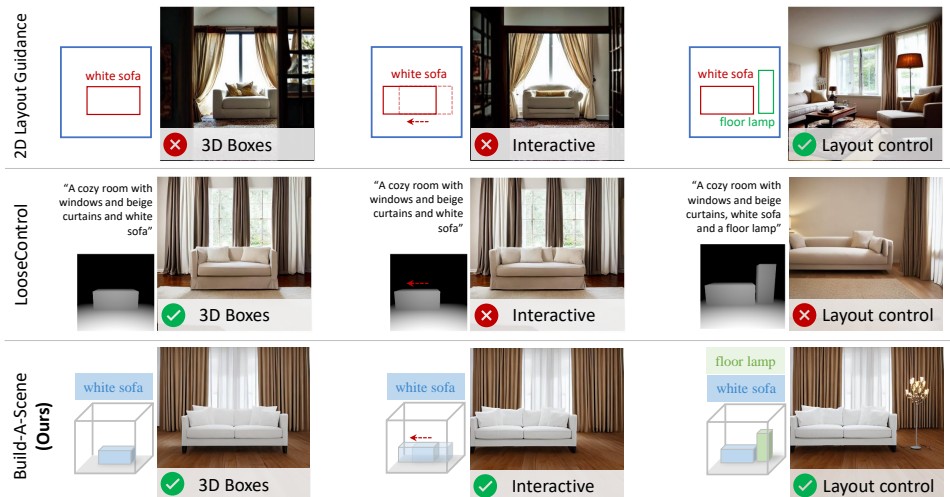

Figure 2: Existing 2D layout control approaches, *i.e.*, Layout Guidance Chen et al. (2024) accept static 2D layouts with no mechanism for 3D control or interactively changing the layout while preserving the image. LooseControl Bhat et al. (2024) can generate images conditioned on 3D boxes but cannot interactively move objects or handle a layout with diverse objects. Build-A-Scene is the first approach to support 3D layouts and allows users to manipulate them interactively.

# 1 INTRODUCTION

Recent advancements in diffusion models (Rombach et al., 2022; Ramesh et al., 2022; Saharia et al., 2022; Podell et al., 2024) have profoundly revolutionized image generation, making it a pivotal task in various creative domains, including design, art, and media production. Image diffusion models excel at generating high-quality images that adhere to a given textual prompt, effectively describing the image contents. This enhances the capability and efficiency of creators by enabling them to generate concepts and designs solely by describing their thoughts in text. However, multiple studies (Chen et al., 2024; Lian et al., 2023; Feng et al., 2023) have shown that diffusion models struggle to follow textual prompts accurately. More specifically, they encounter difficulties following object count, comprehending object placement, and understanding the relationship between objects.

Several approaches (Chen et al., 2024; Lian et al., 2023; Feng et al., 2023; Couairon et al., 2023; Xie et al., 2023; Gani et al., 2024) attempted to alleviate these shortcomings by investigating *2D layout control*. These approaches require the user to provide a layout describing each object's size, shape, and location in the image alongside their respective textual description. Nonetheless, existing approaches for layout control adopt 2D inputs such as points, bounding boxes, or segmentation maps with no mechanism to position objects in 3D. This limits the creators' controllability in applications that require 3D control over the location and orientation (in-plane rotations) of objects, such as interior design and complex scene generation. Moreover, existing approaches for layout control require the user to provide a static layout beforehand and would fail to preserve the generated image under any layout changes, *e.g.*, moving or scaling an object. Figure 2 shows an example of this scenario for the 2D layout approach, Layout-Guidance Chen et al. (2024), where moving the box that contains the sofa leads to changes in the sofa itself and the rest of the scene (the door).

To control the position of objects in 3D, several depth-conditioned diffusion models (Zhang et al., 2023; Mou et al., 2023) have been proposed to generate images for given depth maps. Furthermore, LooseControl (LC) (Bhat et al., 2024) introduced the use of rendered 3D boxes and planes as a conditioning signal for T2I models, enabling control over the location and orientation of objects in 3D. However, LooseControl was not designed to follow a layout with a diverse set of objects and relies solely on textual prompts to describe the image's contents. It is often observed that when dealing with multiple types of objects, some objects are either omitted or placed in an incorrect location, as demonstrated in Figure 2. In addition, any changes to the 3D boxes used as guidance alter the generated objects and might cause artifacts. This limits the usability of LC in generating complex scenes with diverse objects.

We introduce Build-A-Scene, an *interactive* training-free approach for T2I with *3D layout control*. Our approach formulates the image generation process as a multi-stage building process where the user starts with an empty scene and populates it using an interactive 3D layout. We achieve this by leveraging the existing depth conditioning model LC to replace the 2D bounding boxes in layout control with 3D boxes. Furthermore, we propose a Dynamic Self-Attention (DSA) module that allows seamlessly adding objects to a scene while preserving the existing contents. Additionally, we introduce a strategy for consistent 3D translation that preserves the identity of objects under layout changes. To evaluate our approach, we establish a benchmark and an evaluation protocol for the task of interactive 3D layout control. Experiments show that our approach can generate complicated images and outperforms LC by a factor of 2 on object generation success rate and even on adherence to the user-provided 3D boxes. It even outperforms the 2D Layout-Guidance by $\sim 15\%$ on object generation success rate despite being training and guidance-free. Moreover, it outperforms Layout-Guidance and LC in preserving objects under layout changes on all metrics.

## 2 RELATED WORK

In this section, we give a brief overview of existing approaches for 2D layout control. Since we introduce a new strategy for preserving object identity under layout change, we describe existing approaches for the task of consistent object generation in diffusion-based Text-to-Image (T2I).

### 2.1 LAYOUT CONTROL IN T2I DIFFUSION MODELS

The objective of layout control is to allow the user to explicitly specify where each element of the generated image should be placed. Existing approaches employ 2D layouts with various types of annotations, including points, scribbles, bounding boxes, and segmentation masks. These layouts are incorporated into the image generation process either by fine-tuning the pre-trained diffusing models to incorporate them as additional conditions or in a training-free manner. (Yang et al., 2023b; Zheng et al., 2023) trained additional modules to incorporate the layout as coordinates into a pre-trained diffusion model. (Li et al., 2023; Zhou et al., 2024; Nie et al., 2024; Avrahami et al., 2023b) train different modules to condition the image generation process on bounding boxes, dense blobs, and other types of grounding data. SceneComposer (Zeng et al., 2023) introduced different levels of semantic layouts ranging from text to fine segmentation maps by fine-tuning a pre-trained diffusion model on a richly annotated dataset. This category of approaches requires fine-tuning of pre-trained diffusion models, which comes at a computational and data annotation cost.

The training-free approaches attempted to solve the problem in a zero-shot manner to avoid complexities associated with fine-tuning diffusion models. (Xie et al., 2023; Chen et al., 2024; Liu et al., 2024) employed guidance strategies over the cross-attention responses to steer the denoising process in a direction that fulfills the layout specification. For a finer level of layout control, Zest (Couairon et al., 2023) employed layout segmentation maps and utilized a segmentation model to guide the diffusion process to align with the segmentation maps. None of these aforementioned approaches are capable of controlling object location and orientation in a 3D scene, *i.e.*, 3D layout control. Moreover, they expect the user to provide a static layout beforehand and do not offer any mechanisms to change layout elements while preserving the rest of the image. We attempt to tackle these shortcomings by introducing the first interactive 3D layout control approach for T2I.

### 2.2 CONSISTENT OBJECT GENERATION IN T2I

Several approaches have studied the problem of consistent object generation, *i.e.*, personalized generation, in diffusion models to generate consistent variations of a specific object. (Ruiz et al., 2023; Hu et al., 2021; Wang et al., 2023) fine-tunes a pre-trained diffusion model on a set of images of a specific subject, which allows for generating consistent images of the subject as specified by the text prompt. (Ye et al., 2023; Ma et al., 2024; Wang et al., 2024; Song et al., 2023) followed a different approach and trained adapters to condition pre-trained diffusion models on a single image of the subject. (Yuan et al., 2023) finetuned a diffusion model to be conditioned on a rigid transformation matrix describing the pose of the object. These aforementioned approaches are focused on personalized generation given *user-provided images* and textual prompts. In contrast, our approach aims

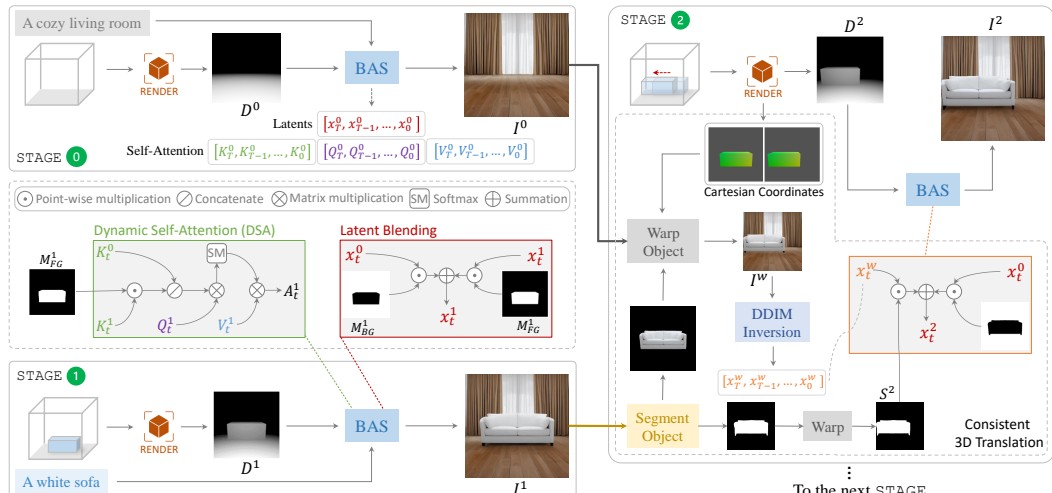

Figure 3: An overview of Build-A-Scene (BAS). We formulate the Text-to-Image (T2I) task as a multi-stage generation process. We illustrate a 3-stage scene. At STAGE 0, the user defines an empty scene with full control over scene size and camera parameters. At STAGE 1, the user adds an object (a white sofa) by defining a box and its corresponding prompt. Our proposed *Dynamic Self-Attention (DSA)* module, coupled with latent blending, ensures that the object is seamlessly integrated into the scene while preserving the existing contents from STAGE 0. At STAGE 2, we illustrate our *Consistent 3D Translation* strategy that allows moving the object in 3D while preserving its identity. Note: *DDIM inversion* is the reverse sampling process of the diffusion model Song et al. (2021), and the *Object Warping* operation is explained in Appendix D.

to generate an image based on a *3D layout* and textual prompts. However, we aim to allow users to change the layout while preserving the generated image, similar to personalized generation.

Another category of training-free approaches for consistent object generation manipulates self-attention to preserve image consistency (Cao et al., 2023; Khachatryan et al., 2023; Qi et al., 2023). The image style can be preserved by injecting the keys and values from a reference image into the self-attention layers of the generated image. However, these approaches are designed to preserve the overall style, but the details of every individual object are not fully preserved (see Figure 4). We propose a novel self-attention module that allows seamlessly inserting objects in an existing scene without altering the existing image contents.

## 3 METHOD

In Text-to-Image (T2I) diffusion models, the objective is to generate an image $I$ given a user-provided textual prompt $P$. In our work, a layout is specified by the user in the form of 3D bounding boxes $\mathcal{B} = \{B^1, B^2, \ldots, B^n\}$ and their corresponding prompts $\mathcal{P} = \{P^1, P^2, \ldots, P^n\}$. Under this setting, the goal is to generate an image where the content enclosed within each bounding box adheres to its respective prompt. More specifically, we have two objectives: 1) Establish an interactive 3D layout pipeline. 2) Ensure object consistency under layout changes.

### 3.1 FROM 2D TO 3D LAYOUT CONTROL

The first step to establishing 3D layout control is finding an appropriate form of 3D annotations that the user can easily create. We leverage LooseControl (Bhat et al., 2024) (LC) for this purpose as it accepts rendered 3D boxes and planes as a conditioning signal in addition to a text prompt $P$. We define the 3D layout as an empty 3D cuboid where the user can add planes to define the boundaries and 3D boxes to define the elements of the scene. We refer to the set of 3D boxes as $\mathcal{B}$ and their corresponding prompts as $\mathcal{P}$ as explained above. Unlike existing 2D layout approaches that require the user to provide the entire layout beforehand, we propose a novel paradigm for layout control by revamping image generation as a sequential process. The user starts with an empty scene and

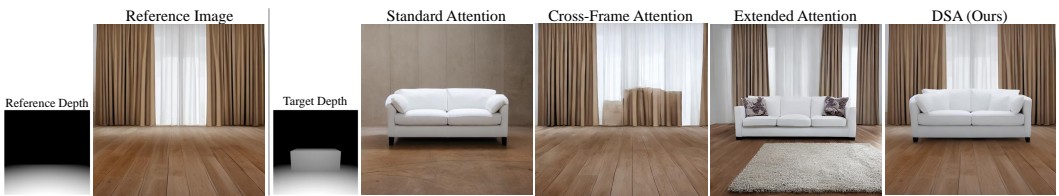

Figure 4: A comparison between existing self-attention mechanisms and our proposed Dynamic Self-Attention (DSA).

interactively adds objects to it through multiple generation stages $i \in [0, n]$. At each stage, the user has control over a single object and can change its type, size, 3D location, and orientation. This greatly enhances user controllability and customizability over the standard layout control. Figure 3 illustrates our pipeline.

At the first stage, *i.e.*, STAGE 0, the background is generated based on an initial prompt that we denote $P^0$ and a rendered depth map of the layout $D^0$ with only planes to define the boundaries. At this stage, the user has full control over the camera model, intrinsic parameters, and viewpoint, providing enhanced 3D control over the generated scene, unlike the existing 2D layout control approaches that lack any control of the camera. At the following generation stages, STAGE $i > 0$, the user adds a 3D box $B^i$ and specifies its corresponding prompt $P^i$. The 3D scene is rendered to obtain a depth map $D^i$, a background and foreground masks for the box being generated at this stage, which we denote as $M_{BG}^i$ and $M_{FG}^i$. During the diffusion process at STAGE $i$, an initial latent code $x_T^i$ is drawn from a random Gaussian noise distribution. DDIM (Song et al., 2021) is used to iteratively denoise the latent code through multiple denoising steps $t : T \to 0$:

$$x_{t-1}^i = \sqrt{\alpha_{t-1}}\, \hat{x}_0^{i,t} + \sqrt{1 - \alpha_{t-1} - \sigma_t^2}\, \epsilon_\theta^t(x_t^i, P^i, D^i) + \sigma_t \epsilon_t \ , \tag{1}$$

$$\hat{x}_0^{i,t} = \frac{x_t^i - \sqrt{1 - \alpha_t}\, \epsilon_\theta^t(x_t^i, P^i, D^i)}{\sqrt{\alpha_t}} \ . \tag{2}$$

where $\alpha_t, \sigma_t$ are the parameters of a noise scheduler, $\epsilon_\theta^t$ is the noise prediction from the diffusion model, and $\epsilon_t$ is random Gaussian noise.

### 3.2 INTERACTIVE 3D LAYOUT CONTROL

At STAGE $i > 0$, we aim to generate a new object based on the 3D box $B^i$ and the textual prompt $P^i$. At the same time, the object is desired to be seamlessly integrated into the scene while preserving the existing contents from previous stages. This can typically be done through inpainting or blended diffusion (Avrahami et al., 2023a). However, they require the user to provide a free-form inpainting mask per object, which is laborious, and they are not directly compatible with the depth-conditioned LC. We propose a novel technique for this purpose that is based on manipulating the self-attention maps and the latent codes.

In standard image diffusion models with a UNet backbone, *i.e.*, Stable Diffusion 1.5 (Rombach et al., 2022), each residual block has self-attention modules, which were found to encode the style and the structure of the generated image (Tumanyan et al., 2023). Self-attention for a given block and timestep $t$ at STAGE $i$ is computed as:

$$A_t^i = \text{Softmax}\left( \frac{Q_t^i {K_t^i}^\top}{\sqrt{d_{k_t^i}}} \right) V_t^i \ , \tag{3}$$

$$Q_t^i = f_t^i\, W_Q, \qquad K_t^i = f_t^i\, W_K, \qquad V^i = f_t^i\, W_V.$$

where $f_t^i$ are the intermediate UNet features, and $W$ are trainable projection matrices.

A widely used approach to transfer the style of a reference image to a target image is *cross-frame attention* (Cao et al., 2023; Khachatryan et al., 2023), which replaces $K^i$ and $V^i$ in Equation (3) with those of the reference image, *i.e.*, $K^{i-1}$ and $V^{i-1}$. This suggests that the target image will query

the reference image for style, resulting in a consistent style between the two images. This approach was adopted in LC, but it was found to be incapable of generating new objects with a different style, as shown in Figure 4. This is intuitive as we limit the target image to copy the style exclusively from the source image. An alternative approach is the extended attention adopted in Qi et al. (2023) for performing consistent video edits, where the target image queries style from multiple images. Figure 4 shows that this strategy can generate objects with a new style but deviates from the overall style of the reference image as different styles are mixed in an uncontrolled manner.

We propose a Dynamic Self-Attention (DSA) technique, which is able to freely generate an object with a new style while preserving the existing elements of the image. We achieve this by augmenting the attention keys to include the keys of STAGE $i-1$ and a masked window of the keys of STAGE $i$:

$$
\begin{aligned}
\hat{K}_t^i &= [K_t^{i-1^\top} \oslash [K_t^{i^\top} \odot M_{FG}^i]] \ , \\
\hat{V}_t^i &= [V_t^{i-1^\top} \oslash [V_t^{i^\top} \odot M_{FG}^i]] \ , \\
A_t^i &= \text{Softmax}\left(\frac{Q_t^i \, \hat{K}_t^i}{\sqrt{d_{k_t^i}}}\right) \, \hat{V}_t^i \ ,
\end{aligned}
\tag{4}
$$

where $\oslash$ is the concatenation operator, and $\odot$ is point-wise product. This enforces the diffusion model to copy the overall style of the previous stage and allows it to generate a new style within the box of the current stage. It is noteworthy that our approach is plug-and-play into the pre-trained diffusion model and does not require any finetuning. Moreover, it does not rely on guidance similar to (Chen et al., 2024), which requires backpropagating through the diffusion model at some iterations, adding a computation overhead.

To enhance the preservation of the background of the previous stage (especially when the latent distribution changes between stages), we blend the latent codes as follows:

$$
x_t^i = M_{BG}^i \, x_t^{i-1} + M_{FG}^i \, x_t^i \ ,
\tag{5}
$$

Similarly, the features can be blended for more preservation. Finally, to harmonize the colors of the scene, we follow (Yang et al., 2023a), and we optionally perform AdaIN (Huang & Belongie, 2017) between $x_t^{i-1}$ and $x_t^i$. We obtain the final image $I^i$ for STAGE $i$ at the end of the denoising process.

### 3.3 CONSISTENT 3D TRANSLATION

A major limitation of existing layout control approaches is their inability to preserve objects under layout changes, *i.e.*, scaling and translation. LC also suffers from the same problem as demonstrated in Figure 2. This was attributed to the distributional shift of the latents that are aligned with the object before and after the layout change (Eldesokey & Wonka, 2024). Therefore, we propose a strategy for preserving objects under layout changes, *i.e.*, 3D translation.

To translate an object at STAGE $i$, we start by segmenting the object out of the generated image in the previous stage $I^{i-1}$. To achieve this without any user intervention, we first obtain a coarse segmentation by accumulating the cross-attention maps that correspond to the object token in $P^{i-1}$ similar to (Cao et al., 2023; Hertz et al., 2022). Then, we fit a bounding box to this coarse segmentation and use it as an input to SAM (Kirillov et al., 2023) to obtain a fine segmentation map $S^{i-1}$ After segmenting the object, we construct a warped image $I^w$ of the object after layout change by pasting the segmented object on the generated image from STAGE $i-2$. To simulate the 3D translation accurately in the image plane, we use the 3D Cartesian coordinates map of the object box before and after the translation to warp the object. More specifically, we compute correspondences for the 4 corners of the objects between the two Cartesian maps and use them to warp the object in the image plane. We also use the same technique to warp the segmentation map $S^{i-1}$ to the new location producing $S^i$. A detailed illustration is provided in the Appendix.

By inverting the warped image through DDIM inversion, we obtain an approximate trajectory $x_T^w, x_{T-1}^w, \ldots, x_0^w$ of the latents corresponding to the object after changing the layout. Finally, we blend the latents between $x_t^w$ and $x_t^{i-2}$ as follows:

$$
x_t^i = S^i \, x_t^w + (1 - S^i) \, x_t^{i-2} \ .
\tag{6}
$$

This blending allows the diffusion model to regenerate the object of interest at the new location while preserving the background. We perform this blending for a number of timesteps $\mathcal{T} <= T$.

| | 3D Layout Control | | | Object Consistency | | | | | RT |
|---|---|---|---|---|---|---|---|---|---|
| | CLIP$_{T2I}$ ↑ | OA ↑ | mIOU ↑ | CLIP$_{I2I}$ ↑ | SSIM ↑ | PSNR ↑ | 3D-Dir ↑ | 3D-Rot ↓ | |
| Layout-Guidance | **0.323** | 48.2 | 0.425 | 0.838 | 0.189 | 28.35 | 0.48 | 29° | 12 s |
| LooseControl | 0.302 | 24.3 | 0.633 | 0.924 | 0.367 | 29.12 | 0.65 | 27° | 2 s |
| **Ours** | 0.321 | **53.5** | **0.772** | 0.924 | **0.463** | **29.3** | **0.88** | **22°** | 6 s |

Table 1: A quantitative comparison between our proposed approach, the 2D layout control method Layout-Guidance Chen et al. (2024), and LooseControl Bhat et al. (2024). RT refers to runtime.

## 4 EXPERIMENTS

In this section, we provide a qualitative and quantitative evaluation of our proposed approach. For clearer insights, we split the evaluation into two sub-tasks: (1) *3D layout control* and (2) *object consistency under layout change*.

### 4.1 EXPERIMENTAL SETUP

**Comparison** We compare against the baseline LooseControl (LC) Bhat et al. (2024) that we employ for depth conditioning in our pipeline. To show where our approach stands with respect to 2D layout approaches, we also compare against Layout-Guidance Chen et al. (2024) that accepts 2D bounding boxes. To map the 3D boxes to 2D, we fit a bounding box to box masks $M_{FG}^i$.

**Implementation Details** LC is based on ControlNet with Stable Diffusion v1.5 Rombach et al. (2022) that is fine-tuned through LoRA adaptation Hu et al. (2021). We keep all the original settings of LC except for the sampler that we changed to a DDIM sampler with a linear schedule as we noticed that it is more stable. We perform $T = 20$ denoising steps in the quantitative comparison for efficiency and $T = 40$ for the qualitative results for better quality. For Layout-Guidance, we use the official implementation with the default parameters. The source code and the evaluation protocol are publicly available. [1]

### 4.2 3D LAYOUT CONTROL EVALUATION

In this task, given a set of 3D bounding boxes and their corresponding prompts, the goal is to generate an image that conforms to these inputs.

**Evaluation Strategy** Since there exist no criteria for evaluating 3D layout control, we define a new evaluation protocol for 3D layout control inspired by its 2D counterpart (Chen et al., 2024; Xie et al., 2023). We define a set of 16 objects from the MS COCO dataset (Lin et al., 2014) and their corresponding aspect ratios. In addition, we define 10 different prompts for diverse scenes such as desert, snow, and room to use for the initial prompt $P^0$. We start by sampling a random scene, object, and create a 3D box that matches the object's aspect ratio at a random z-coordinate. Then, we sample another object, and we randomly select one of the three placements ["left", "right", "above] relative to the first object. We place the second object into the scene similarly, ensuring that it does not occlude the first object or go out of bounds.

We sampled 100 random layouts and ran each layout with 5 different seeds for fairness. Since the baseline LC does not accept layouts, we automatically create a textual description $P^*$ of the scene in the form:

$P^*$: "[$P^0$] with [$P^1$] on the left/right and [$P^2$]on the right/left"

Similarly, we create the textual description for the relation "above" as well.

**Evaluation Metrics** We are interested in evaluating three aspects: how the generated image confronts to the textual description of the scene, whether all specified objects in the layout have been generated, and how well each object fits within its box.

$CLIP_{T2I}$: We compute the CLIP score (Radford et al., 2021) between the final generated image and the textual description of the image $P^*$.

---

[1]https://github.com/abdo-eldesokey/build-a-scene

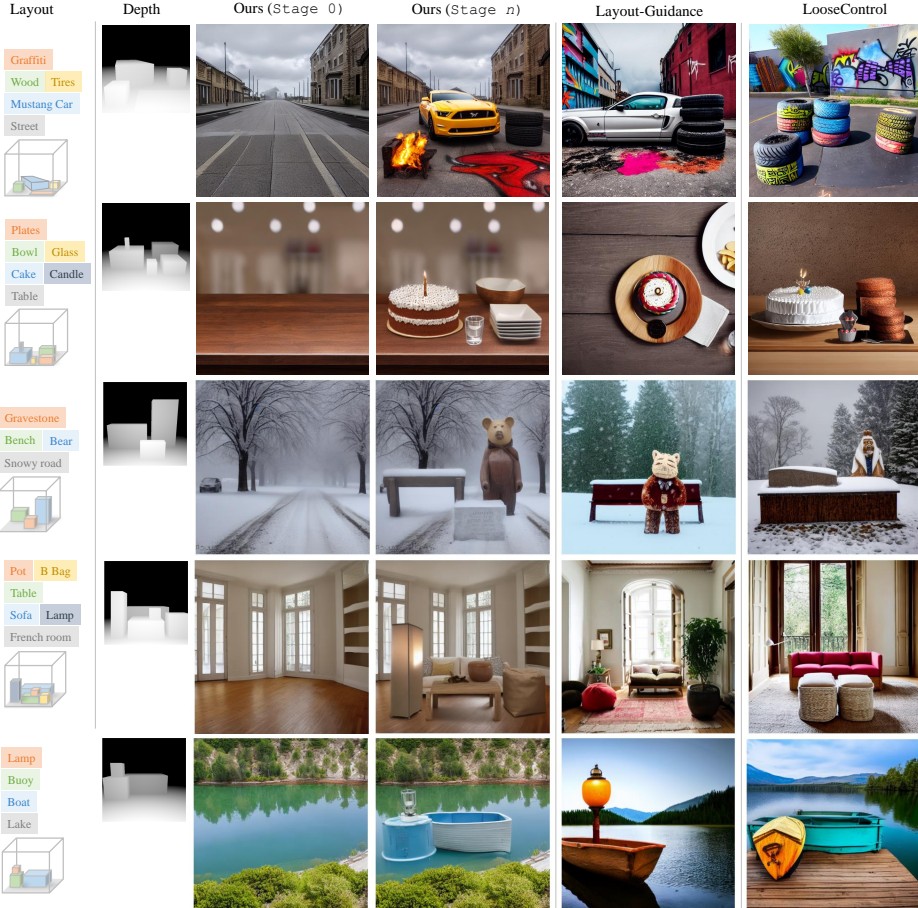

Figure 5: A qualitative comparison on the task 3D layout control.

*Object Accuracy (OA):* We use a general object detector, YOLOv8 (Reis et al., 2023), to check if objects specified by the layout are detected in the image.

*Mean Intersection-over-Union (mIoU):* We compute the intersection between the bounding box predicted by YOLOv8 and a bounding box fitted to the 3D box in the image plane. This tells how well the object is enclosed within the layout box.

**Quantitative Results** Table 1 summarizes the quantitative comparison. Our approach scores two times higher than LC and 15% higher than Layout-Guidance on Object Accuracy (OA), demonstrating its effectiveness in executing the layout. Our approach also outperforms LC and is on par with Layout-Guidance on $CLIP_{T2I}$, demonstrating that the generated image conforms better to the textual description. For the mIOU metric, our approach outperforms Layout-Guidance by a huge margin despite not incorporating any guidance-based techniques. Surprisingly, it also outperforms LC, despite the fact that we have not tuned it. We believe this improvement is caused by our dynamic self-attention that forces the generated objects to lie within their respective boxes.

**Qualitative Results** We provide a qualitative comparison in Figure 5. The figure shows that our approach is more faithful to the layout compared to Layout-Guidance, while LC struggles to generate all objects. We show both STAGE 0 and STAGE $n$ of our approach to highlight how objects are seamlessly integrated into the scene between stages (notice reflections and shadows). We also provide some examples with advanced 3D control of camera and objects in Figure 6.

## 4.3 OBJECT CONSISTENCY UNDER LAYOUT CHANGE EVALUATION

This task aims to move or scale an object in the provided layout while preserving its identity.

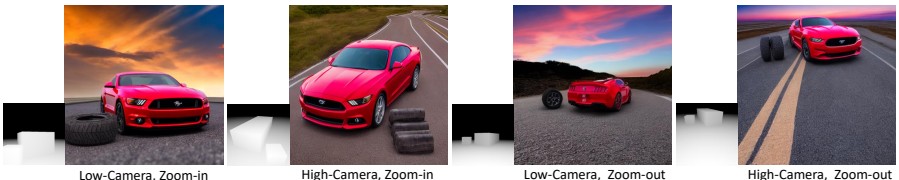

Low-Camera, Zoom-in    High-Camera, Zoom-in    Low-Camera, Zoom-out    High-Camera, Zoom-out

Figure 6: Examples of the advanced 3D control over camera and object provided by our approach.

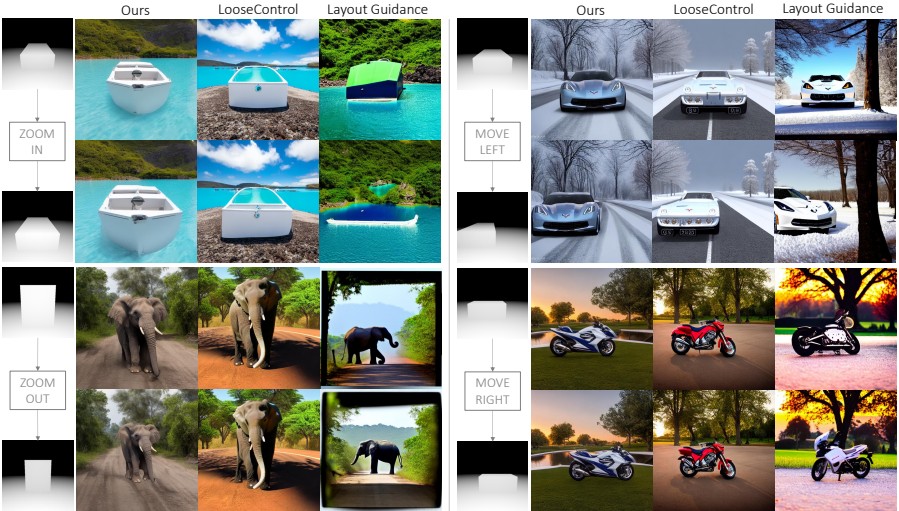

Figure 7: A qualitative comparison for object consistency under layout change.

**Evaluation Criteria** We randomly sampled one of the 16 MS COCO objects and placed it in a layout. Then we randomly selected one of the actions ["move left", "move right", " zoom-in", "zoom-out"]. We ensure that applying the action does not cause the object to be out of bounds. Finally, we compare the object's visual appearance before and after applying the action.

**Evaluation Metrics** Our goal is to evaluate how similar the object is before and after applying some layout action. We crop the object from the images and resize the cropped images to ensure they match. We compute the CLIP score *CLIP$_{I2I}$*, *Structural Similarity (SSIM)*, and *Peak-Signal-to-Noise-Ration Similarity (PSNR)* between a cropped image of the object before and after the action. We also propose *3D Adherence* metrics that evaluate if the generated image adheres to the 3D layout change (see Appendix C for details).

**Quantitative Results** Table 1 summarizes the results averaged over 5 seeds. Our method outperforms other methods on all metrics, demonstrating better preservation of objects and 3D adherence to the layout changes.

**Qualitative Results** We show some qualitative examples for layout changes in Figure 7. For the scaling layout changes, *i.e.*, zoom-in and zoom-out, our approach applies them successfully and seamlessly inserts the object at the new location. LC either distorts the object or does not apply the layout change, while Layout-Guidance changes the object completely. When moving the main object to the left or the right, our approach successfully applies the changes and preserves the object. On the other hand, LC distorts the object, while Layout-Guidance changes the object pose.

### 4.4 ABLATION ANALYSIS

Figure 8 provides an ablation analysis for the impact of different components of our pipeline. When the Dynamic Self-Attention (DSA) is disabled, the model is not capable of inserting a new object into the reference image. Skipping the latent blending in Equation (5) causes some details of the background to change (the paintings on the wall). Using AdaIN (Huang & Belongie, 2017) contributes to harmonizing the colors of the generated object with the background.

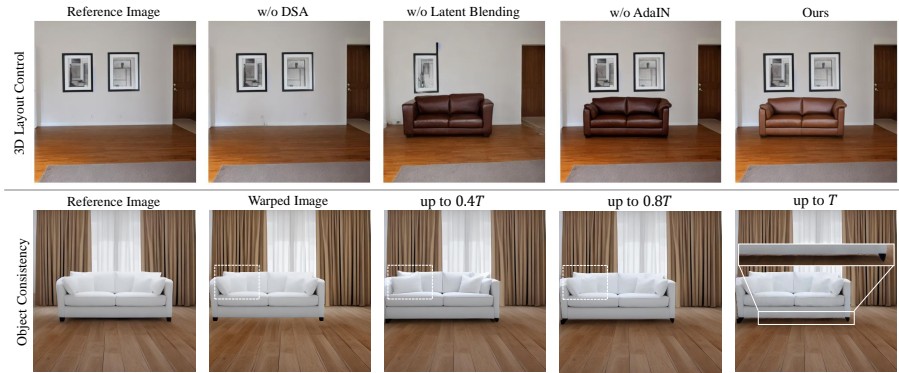

Figure 8: Ablation analysis for different parts of our pipeline.

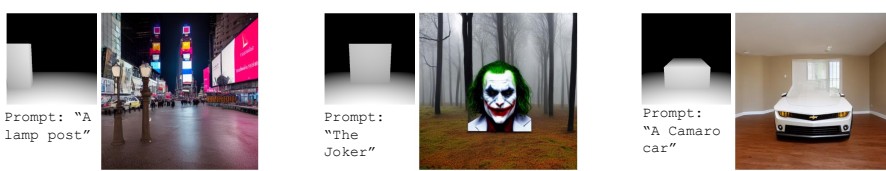

Figure 9: Limitations of our approach.

We also experiment with varying $\mathcal{T}$ in Section 3.3 for blending the latents. The warped image lacks realism, and the sofa appears to be floating. By applying Equation (6) for $\mathcal{T} = 0.4T$, the sofa is seamlessly integrated into the scene, but some of the details (the cushions) are not perfectly constructed. When $\mathcal{T} = 0.8T$, the sofa is seamlessly blended into the scene, and the fine details are well constructed. With $\mathcal{T} = T$, some artifacts start to appear at the boundaries.

## 5 LIMITATIONS AND FUTURE WORK

Figure 9 shows some of the limitations of our approach. First, our approach is sensitive to the aspect ratio of the box. If provided with a box wider than the actual width of the object, it can generate two instances of the object (left figure). When the aspect ratio of the box is not suitable for the specified prompt, it can generate a distorted object or a photo of the object (middle figure) We believe that this is a natural behavior of our approach as it tries to fulfill the layout and the prompt requirements concurrently. Secondly, if a large object is placed in a small space, *i.e.* a box intersecting with the boundaries, such as a car in a room, the out-of-boundary parts of the car are distorted (right figure). In general, the definition of boxes needs to be reasonable to obtain the desired results.

For the Consistent 3D Translation strategy, if the object segmentation part fails, it becomes infeasible to preserve the objects. Finally, one might argue that the multi-stage generation pipeline adds a computational overhead to the generation process. However, this is a fair price in return for the enhanced control over scene elements that helps the user reach the desired output faster and eventually save time. Moreover, our approach takes 2 seconds per stage and can generate a layout of 5 objects in the same time as Layout-Guidance (see Table 1).

**Future Work** We would like to investigate automated layout generation through a large-language model (LLM) as in (Feng et al., 2023; Gani et al., 2024). Another direction is supporting in-plane rotations while preserving object identity, similar to (Yuan et al., 2023).

## 6 CONCLUSION

We presented a first approach for interactive 3D layout control based on a pre-trained T2I diffusion model. Our approach reformulated image generation as a multi-stage process, providing users with enhanced control over individual objects in 3D. Moreover, we provided the first strategy to preserve objects under layout changes. Experiments show that our approach outperformed both LooseControl and the recent 2D layout control, both quantitatively and qualitatively. We hope that our approach will establish a new research direction for 3D layout control and consistency in layout control.

## 7 ACKNOWLEDGEMENT

The work is supported by funding from KAUST - Center of Excellence for Generative AI, under award number 5940, and the NTGC-AI program.

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

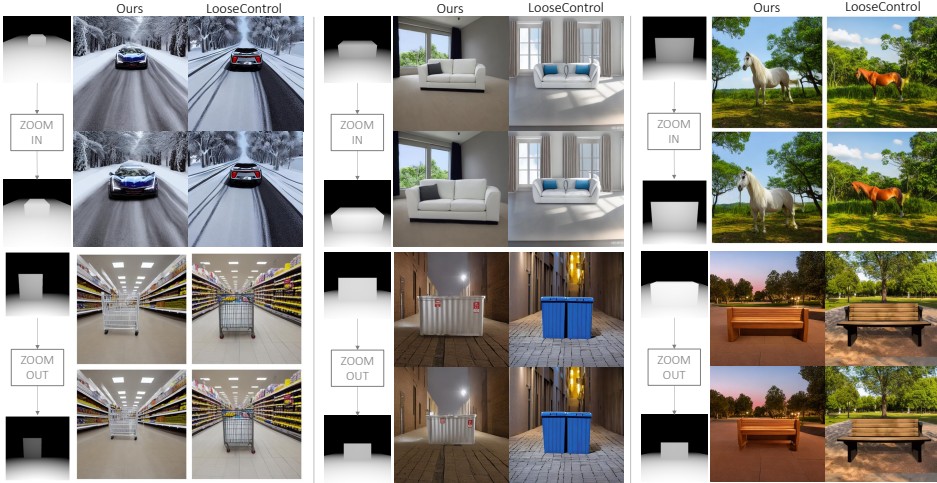

Figure 10: Additional results for our consistent layout change under zoom-in/out layout changes.

## A    APPENDIX

## B    ADDITIONAL QUALITATIVE RESULTS

*Video results* of the interactive generation process and more qualitative results are provided in the *attached supplementary*.

We also provide additional qualitative results for moving objects towards and away from the observer in Figure 10 (zoom-in and zoom-out).

## C    3D ADHERENCE METRICS

We propose two metrics to assess the adherence of different approaches to the requested 3D layout changes. The first metric, which we term **3D-Dir**, evaluates whether the moved box is relocated in the correct direction. Given a layout change to a box in image $I_1$, we store the origin of the box that corresponds to that object before and after applying the layout change as $o_1$ and $o_2$, respectively. After generating a new image $I_2$ that corresponds to the layout change, we use a monocular depth estimation model, *i.e.* Depth-Anything Yang et al. (2024), to obtain depth maps for both images that we denote as $D_1$ and $D_2$. Then, based on the depth maps, we obtain a predicted center point of the object before and after the layout change that we denote as $p_1$ and $p_2$. We compute this with the help of the 3D box masks $M_{FG}^1$ and $M_{FG}^2$ before and after the layout change. Afterward, we define two directional vectors $x = o1 - o2$, and $y = p_1 - p2$ as reference and predicted trajectories. Finally, we compute the cosine similarity between these two vectors to obtain the 3D Adherence score:

$$3D\ Adherence = \frac{x \cdot y}{\|x\|\|y\|}, \tag{7}$$

A high score indicates that the reference and predicted vectors have a similar direction and magnitude, meaning the desired layout change was executed correctly.

The second metric is **3D-Rot**, which examines if the orientation of the generated object before and after the layout change aligns with the 3D box. For this purpose, we employ the Omni3D Brazil et al. (2023) detector that predicts 3D bounding boxes for objects in images $I_1$ and $I_2$. The Omni3D detector also predicts the pose of the box in 6D representation that we transform to Euler angles for comparison. We compute the difference between the Euler angles of the 3D box and the predicted 3D bounding box from $I_1$ and $I_2$ to obtain differences $\theta_1^{\text{yaw}}, \theta_1^{\text{pitch}}, \theta_1^{\text{roll}}$ and $\theta_2^{\text{yaw}}, \theta_2^{\text{pitch}}, \theta_2^{\text{roll}}$. Finally, we average over all these angle differences to obtain an average deviation in degrees. Our method scores the best on these two metrics, and we provide the results Table 1. It is worth noting that Omni3D is less accurate than 2D object detectors, which leads to inflated error rates across all methods. Our approach achieves the lowest error relative to the other methods in comparison.

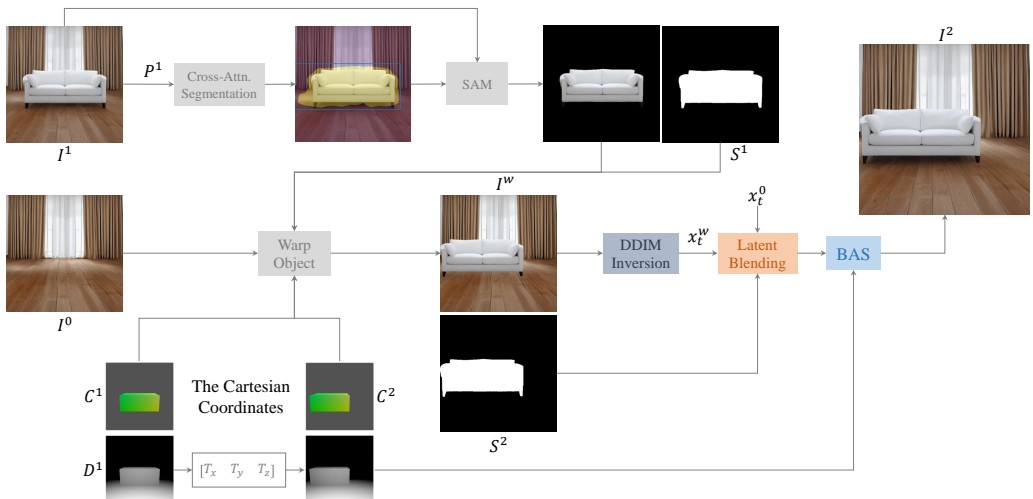

Figure 11: A detailed illustration of our Consistent 3D Translation strategy.

## D  OBJECT WARPING

We provide a more detailed explanation of our 3D Consistent 3D Translation in Figure 11. To translate an object at STAGE $i$ with vector $\hat{T}$, we start by segmenting out the object based on the cross-attention maps of the object prompt $P^{i-1}$ to provide a coarse segmentation map (shown in yellow). We fit a bounding box (shown in blue) to the coarse segmentation map, and we use it as an input prompt to the SAM segmentation model to provide a fine segmentation map $S^{i-1}$. Given the segmented object, we aim to paste it on the image from STAGE $i-2$ at the new location. However, since we apply the translations in 3D, there is a perspective change that can not be captured by directly cutting and pasting the object. Instead, we use the Cartesian coordinates of the rendered 3D box to warp the segmented object and the fine segmentation map to the new location in the warped image. To achieve this, We render the object box using ray-tracing before and after the 3D translation to obtain depth maps $D^{i-1}, D^i$ and Cartesian coordinates maps $C^{i-1}, C^i$ for the point-of-hit of the rays. We align $C^i$ with $C^{i-1}$ by subtracting the translation vector $\hat{T}$ from the former. Finally, we warp the 4 corners of the segmented object based on the Cartesian coordinate maps to produce the warped image $I^w$, and warped mask $S^2$. This strategy ensures that the object follows the perspective change of the 3D box and scales the object according to the translation in 3D, not the image plane.

Given the warped image $I^w$, we invert it using DDIM to obtain a trajectory of latents $x_t^w$. Then, we generate the final translated image $I^2$ using latent blending based on Equation (6).

## E  MULTI-STAGE GENERATION

To illustrate how our approach maintains scene consistency across different stages, we visualize individual stages for several examples in Fig. 12. The figure clearly showcases the effectiveness of our proposed Dynamic Self-Attention (DSA) module in seamlessly integrating new objects into the scene while preserving the original elements.

## F  EVALUATION PROTOCOL

We present the details for the benchmark and evaluation protocol that was used to evaluate our approach in Figure 13. First, we randomly select a scene from the 'scenes' list. For every scene, there is a pre-defined list of objects that naturally appear in this scene. Based on this pre-defined list, we select objects to be added to the scene. To create a reasonable 3D box for every object, the aspect ratios for different objects were generated by prompting ChatGPT, and then manually refining them.

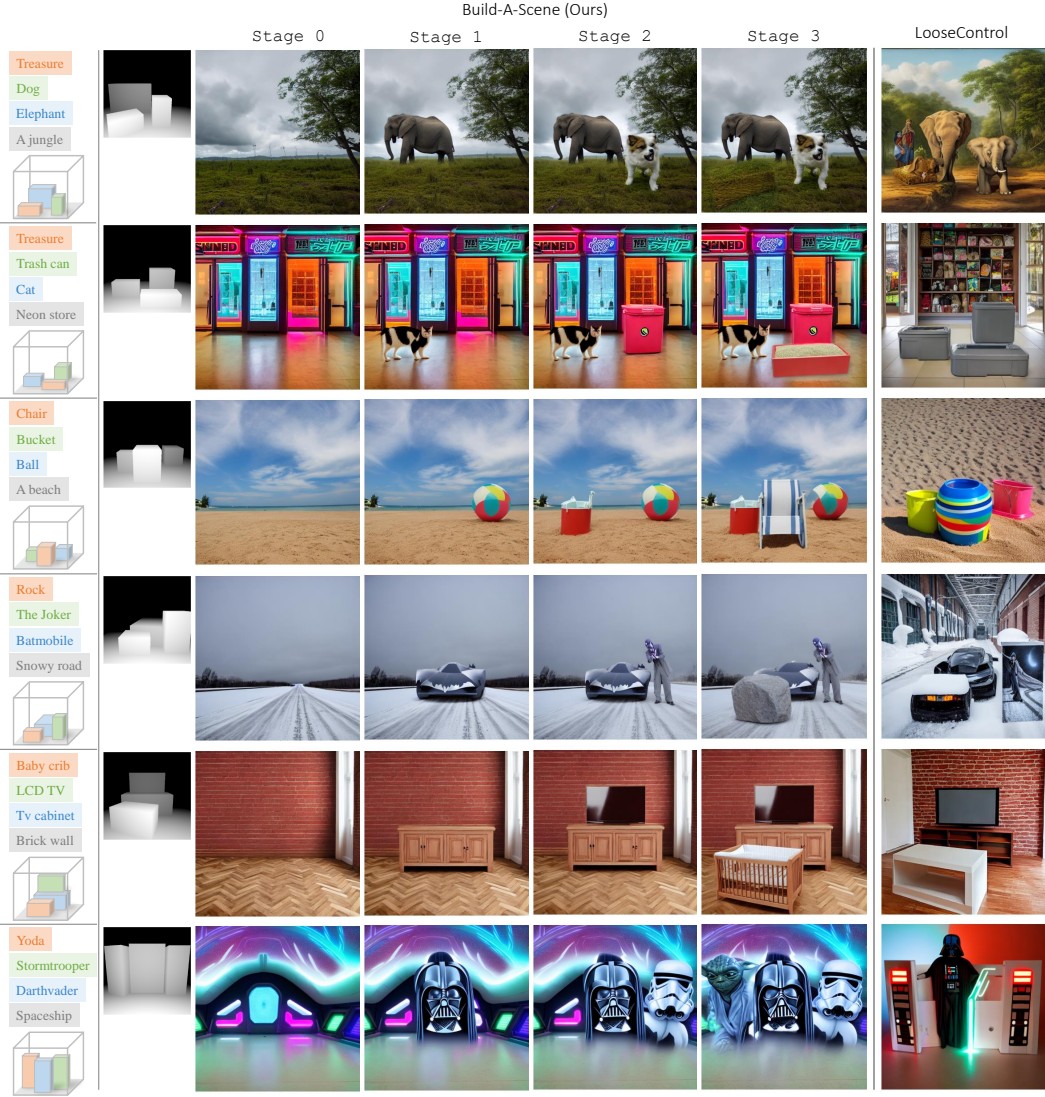

Figure 12: A demonstration for the effectiveness of our proposed Dynamic Self-Attention (DSA) in seamlessly inserting objects while preserving existing ones.

```python
object_categories = {
    "animals": ["a cat", "a dog", "a horse", "an elephant", "a grizzly bear"],
    "indoor": ["a teddy bear", "a microwave", "a backpack", "an lcd tv", "a sofa", "a
chair", "a table", "a bed"],
    "outdoor": ["a car", "a motorcycle", "a backpack", "a bench", "a sofa"],
}

scenes = [
    ("An empty desert with cloudy sky", ["animals", "outdoor"]),
    ("An empty room with windows and curtains", ["indoor"]),
    ("An empty street", ["outdoor"]),
    ("An empty jungle", ["animals"]),
    ("An empty road", ["animals", "outdoor"]),
    ("An empty studio", ["indoor"]),
    ("An empty beach", ["animals"]),
    ("A snowy landscape", ["outdoor"]),
    ("An empty apartment", ["indoor"]),
]

aspect_ratios = { # Width, Depth, Height
    "a cat": (0.2, 0.2, 0.4),
    "a dog": (0.3, 0.2, 0.5),
    "a horse": (0.7, 0.3, 0.5),
    "an elephant": (0.9, 0.3, 0.6),
    "a grizzly bear": (0.8, 0.3, 0.5),
    "a teddy bear": (0.3, 0.2, 0.4),
    "a microwave": (0.4, 0.2, 0.25),
    "a backpack": (0.3, 0.1, 0.4),
    "a car": (0.6, 1.2, 0.4),
    "a motorcycle": (0.8, 0.2, 0.35),
    "an lcd tv": (0.5, 0.05, 0.3),
    "a sofa": (0.9, 0.3, 0.4),
    "a chair": (0.25, 0.25, 0.5),
    "a table": (0.5, 0.5, 0.4),
    "a bench": (0.7, 0.3, 0.3),
    "a bed": (0.6, 0.7, 0.25),
}

relations = {
    "a cat": ("l", "r", "a"),
    "a dog": ("l", "r", "a"),
    "a horse": ("l", "r"),
    "an elephant": ("l", "r"),
    "a grizzly bear": ("l", "r"),
    "a teddy bear": ("l", "r", "a"),
    "a microwave": ("l", "r", "a"),
    "a backpack": ("l", "r", "a"),
    "a car": ("l", "r"),
    "a motorcycle": ("l", "r"),
    "an lcd tv": ("l", "r", "a"),
    "a sofa": ("l", "r"),
    "a chair": ("l", "r"),
    "a table": ("l", "r"),
    "a bench": ("l", "r"),
    "a bed": ("l", "r"),
}
```

Figure 13: Objects and scenes used for the evaluation Protocol, "l", "r", and "a" in *relations* stand for left, right, and above respectively.

The relationships between objects were selected based on how they usually co-occur in nature. For instance, an LCD TV can be on top of another object, but an elephant can not.

## G  USER INTERFACE

To enable interactive scene building with our approach, we developed a Gradio-based web interface that is illustrated in Figure 14. At each stage, users can create a box and manipulate its scale,

Figure 14: The user interface for Build-A-Scene (BAS)

translation, and rotation in 3D space. The box's contents are defined through a text prompt, and clicking the "Generate" button produces the corresponding image. Users can then iteratively refine the box, modify the prompt, and adjust the seed to achieve the desired outcome. Once satisfied with the current stage, users can click the "Next Stage" button to save the scene's state and proceed. A "Start Over" button is also provided to create a new scene from scratch.

