# OpenReview forum: "Build-A-Scene: Interactive 3D Layout Control for Diffusion-Based Image Generation"
_ICLR.cc/2025/Conference — ICLR 2025 Poster_

### Official Review · Reviewer_X2dB · 2024-10-18

**Soundness:** 2
**Presentation:** 2
**Contribution:** 2
**Rating:** 6
**Confidence:** 3

**Summary:**

This work is trying to generate an image based on the layout of 3D boxes for each object we desire in the image. It builds upon a depth-conditioned stable diffusion generator, and they make it an iterable fashion where they focus on different object placement in each step to enhance control of each separate object. to keep the information from the last iteration to the next, they presented the Dynamic Self-Attention (DSA) module which blends between attention maps from the previous iteration and the new one, specifically on the mask where the new object is placed.

**Strengths:**

they present an interesting 3D layout control over generated images, Furthermore, the multi-stage to have more control over each object sounds good.

**Weaknesses:**

1. you presented in most of the paper examples of moving the object aside (Fig.2), however, with 3D boxes, it is much more interesting to see examples of moving toward me or away from the observer. you provided only two examples like this in Fig.6, I would love to see it as the focus of the paper.
2. your evaluation metrics (Table. 1) have no comparison to 3D information, which is the focus of the paper. i would love o see a new metric that can evaluate the 3D placement, like with monocular depth estimation, or 3D object detection networks.

**Questions:**

from your pipeline figure (Fig.3) it seems like you didn't talk about some key parts of the method, specifically in stage 2 (warping, DDIM inversion) it might not be novel, but helps to grasp the notion of the method.

---

> ### Author Response · Authors · 2024-11-24
>
> 1- Moving objects forward and backward is definitely essential. Therefore, we provided more examples of moving objects toward and away from the observer in the appendix of the revised version of the paper (please see Figure 10 in the revised PDF).
>
> 2- That is a great suggestion! We propose a depth-based metric to evaluate the adherence to the 3D layout change. We denote this new metric as *3D Adherence*, and we add it in Table 1 in the revised version of the paper. The metric computes the cosine similarity between the translation vector requested by the user, and the predicted translation vector based on a monocular depth estimation such as Depth-Anything.  We provide a detailed explanation of the metric in Appendix C in the revised version of the paper.
>
> **DDIM and Warping** We provided a detailed explanation of the object warping in Appendix C, and we added a brief description of DDIM in the caption of Figure 3.

---

> > ### Comment · Reviewer_X2dB · 2024-11-25
> >
> > An interesting choice for the 3D metric?
> > Why was this metric chosen?
> > What about the rotation of the bbox?

---

> > > ### Author Response · Authors · 2024-11-25
> > >
> > > Since we generate images with various object classes, employing a 3D object detector is infeasible, as no existing 3D object detector supports a wide range of objects. Most 3D object detectors are specialized for specific domains, such as driving scenarios, and lack generalizability to diverse object categories.
> > > Therefore, employing a monocular depth estimation for evaluation was the only viable option as it is *object-agnostic*.
> > >
> > > The primary focus of Section 4.3 is on evaluating *object consistency under layout changes*, and we assess this consistency through two key aspects:
> > > 1. *Object Appearance Consistency*: Measured using CLIP$_{I2I}$, PSNR, and SSIM.
> > > 2. *Adherence to the Layout Change*: Evaluated through the 3D Adherence metric.
> > >
> > > Regarding the rotation of bounding boxes, while it is possible to incorporate this into the evaluation, it does not contribute to assessing consistency under layout changes and would complicate the metric further.

---

> > > > ### Comment · Reviewer_X2dB · 2024-11-25
> > > >
> > > > i raised my rating for the new metric
> > > > i am willing to raise my rate more if more metrics are added
> > > > 1. 3D bbox for the objects that have 3D detector (like the car, ...)
> > > > 2. the 3D Adherence metric will include the rotation as well

---

> > > > > ### Author Response · Authors · 2024-11-27
> > > > >
> > > > > Thank you for your response and for raising the score!
> > > > >
> > > > > For the additional metrics, we conducted a comprehensive search of 3D object detectors, and we found a model, Omni3D [1], that trains on both indoor and outdoor images, making it ideal for our evaluation.
> > > > >
> > > > > Therefore, we proposed another metric to evaluate the 3D adherence of rotation, which we denoted as 3D-Rot.
> > > > > Please see the revised PDF for details (Table 1 and Appendix C).
> > > > >
> > > > > [1] Brazil, Garrick, et al. "Omni3d: A large benchmark and model for 3d object detection in the wild." Proceedings of the IEEE/CVF conference on computer vision and pattern recognition. 2023.

---

### Official Review · Reviewer_JbbB · 2024-11-01

**Soundness:** 3
**Presentation:** 3
**Contribution:** 2
**Rating:** 8
**Confidence:** 3

**Summary:**

This paper proposes a interactive 3D layout control approach for T2I based on diffusion approach.  Beginning with an generated scene images without objects inside, the proposed approach allows uses to add 3D box and its text description in the bounding box of the scene to specify the 3D layout of the scene.  The depth map of the 3D layout and the text description are used as the condition to guide diffusion model, and the keys are combined according to the mask of the box to maintain the style of the image in each stage, denoted as dynamic self-attention.

The video demo is nice, which demonstrates the 3D layout control process clearly.

**Strengths:**

1. A straightforward yet effective approach to control the scene layout and the styles of objects placed in the scene.  With the 3D interface, the user can control how to put the objects on other objects or the relative position between objects, more natural than 2D interface.  The ability to move the object closer or farther away from the camera is also attractive.

2. The DSA module designed to merge the keys at different stage is easy to implement, and it allows to generate new object of a different style. Also, the combination of latent codes according to the mask  and AdaIN operation can effectively maintain the image styles surrounding the newly inserted object.

**Weaknesses:**

The composes scene is relatively simple.  All objects are put on the flat surface, such as floor or desk top. It might be partially attributed to the box-based 3D layout control interface.  For example, can we put an object on the sofa if we use freeform surface as the control interface?  If freeform surface, how much effort does the user should pay to place a new object such that the diffusion model can generate the image correctly?

**Questions:**

In the DSA model,  why do you only combine the key tokes between stage i-1 and stage i?  I am wondering the effect of combining V tokens instead.  Discussions on why chose key tokens in DSA is necessary for the deep undstanding the design of DSA.

---

> ### Author Response · Authors · 2024-11-24
>
> We thank the reviewer for the positive feedback and the constructive comments. We address these comments below.
>
> *Freeform Surfaces* We agree with the reviewer’s observation; placing objects on freeform surfaces, such as a sofa, can be challenging due to the box-based interface. For instance, placing a cat on a sofa would require multiple iterations to align the boxes so the cat is placed on the seat of the sofa. A potential solution to this problem is overlaying the layout on the generated image so that the user can get an estimate of where the new box can be placed.
>
> *The use of V tokens in DSA* Thank you for pointing this out! In fact, a line was missing in equation (4), which concatenates the V tokens from stages i and i-1 as well. This is necessary so that the dimensionality of the keys and values match. We fixed this mistake in the revised version of the paper.

---

### Official Review · Reviewer_oNnn · 2024-11-02

**Soundness:** 3
**Presentation:** 2
**Contribution:** 2
**Rating:** 3
**Confidence:** 5

**Summary:**

BAS introduces a pipeline which allows users to guide the image synthesis through interacting with boxes in the 3D space and using the text prompt. The boxes reflect the position of objects in the image. To keep the image consistency throughout interacting, the author propose the dynamic self attention and object wrapping in order to maintain the background as much as possible. As a new work to introduce 3D interacting into 2D image synthesis, it proposes several benchmarks to evaluate its performance and compare to some image synthesis works.

**Strengths:**

Firstly, using inpainting to preserve the background during the interacting is a simple but sound idea, and the manipulations in 3D world make the masks’ calculation fully under control. Furthermore, 3D guidance is common and useful when we want to show the relative transformation of objects to the diffusion model, as we usually want it to give image of the real 3D world.

**Weaknesses:**

(1) The technical contribution seems not to be so sufficient. As the T2I techniques and pre-trained models are mature, would organizing the masks according to the boxes 3D transformation being sufficient for a paper? Well, the dynamic self-attention block is surely a contribution.
(2) However, in the ablation study, it says “when DSA is disabled, the model isn’t capable of inserting new object”, but in the method, DSA is proposed for maintaining the background, which is different from what may happen in the ablation study. The technical contribution turns out to be confusing and doubtful.
(3) Another consideration is that your run time is 3 times of “loose control”. Would that be harmful to the interaction? Could you please discuss more on that?
(4) Some expression in the “METHOD” still tells why other methods aren’t suitable for this “attention”. That should not appear here.

**Questions:**

No

---

> ### Author Response · Authors · 2024-11-24
>
> We thank the reviewer for the comments and questions. We address them below.
>
> 1- Our paper presents the first method for **interactive 3D layout control**, offering two key contributions:
>
> (a) *3D Layout Control with Greater Freedom*: Our approach is the first to enable users to precisely control the position and orientation of objects and the camera in a 3D space, providing significantly greater flexibility compared to existing layout control techniques.
>
> (b) *Interactive Layout Control*: We introduce the first interactive layout control method that allows users to adjust objects independently without affecting other elements in the image. This saves users a great amount of time iterating over the layout to get the desired outcome and aligns more closely with the natural workflow of artistic creation.
>
> Overall, we provide new and important content creation capabilities.
>
> 2-We state at L270 in the main paper when we introduce DSA, "We propose a Dynamic Self-Attention (DSA) technique, which is able to freely **generate an object** with a new style while **preserving the existing elements of the image**" Therefore, there is no contradiction between the method and ablation sections. DSA has two functionalities that complement each other: (a) inserting a new object into the scene and (b) preserving the background. Please let us know if there is a misunderstanding of the comment.
>
> 3- The time overhead in our approach arises from splitting the image generation process into multiple stages rather than any inherent limitation of the method. In fact, this staged process provides users with greater control over individual objects, enabling them to achieve their desired outcome *faster* than standard layout control approaches. Conversely, approaches like LooseControl cannot work according to layouts with diverse objects, as illustrated in Figure 2, making it unsuitable for this task.
>
> 4- We provided this comparison in the method section to motivate the design of our DSA attention. However, we are open to suggestions from the reviewer on alternative placements of this section.

---

### Official Review · Reviewer_9MBn · 2024-11-04

**Soundness:** 3
**Presentation:** 4
**Contribution:** 3
**Rating:** 6
**Confidence:** 4

**Summary:**

Based on the recent advancements in depth-conditioned T2I, this work presents a novel approach for interactive 3D layout control.     Further, the authors propose a Dynamic Self-Attention (DSA) module and a consistent 3D object translation strategy, to preserve the existing contents and consistent 3D translation.

**Strengths:**

1. The task is well-defined.
2. The model is designed reasonably.
3. The paper is well-written and clearly states the contribution.
4. The results are well-organized, and the experiments are comprehensive.

**Weaknesses:**

1. 3D Awareness. Existing results are mainly about the 3D layout conditioned generation. However, the 3D information is not adequately used. There are several other degrees of freedom in 3D spaces, such as the camera view, the rotation of objects, and the zoom-in/out. These results (even parts) could strengthen this work a lot.

2. User interface. What exactly is the interface? Can you describe the tools for users to use? The interface for the creator is critical, especially for 3D editing. A reasonably designed interface can make this work for real-world applications. Just a demo-based (actually controlled by codes) can not make it practical.

3. The motivation. The motivations of the design of the pipeline is not very clear. The key feature of this work is that it can implement 3D layout-based generation. However, what are the challenges in this task, and why authors must use the proposed pipelines and modules to deal with this challenge?

**Questions:**

See Weakness.

---

> ### Author Response · Authors · 2024-11-24
>
> We thank the reviewer for the feedback and the constructive comments. We address these comments below.
>
> 1- We agree with the reviewer that we did not highlight these aspects sufficiently, and we acknowledge the importance of showcasing them to expose the full potential of our approach. Therefore, we include the following complementary text in Section 3.1:
> “At this stage, the user has full control over the camera model, intrinsic parameters, and viewpoint, providing enhanced 3D control over the generated scene, unlike the existing 2D layout control approaches that lack any control of the camera.”
>
> We also added an additional figure in the main paper (Figure 6) designated for highlighting camera control (high-camera, low-camera), object in-plane rotation, and zoom-in/out (please see the revised PDF).
>
> 2- We developed a user-friendly Gradio user interface that allows users to design each stage interactively.
> We added a screenshot and a description of the interface in Appendix F (please see the revised PDF)
>
> 3- As explained in Section 1 and illustrated in Figure 2, existing approaches for layout control face two significant limitations:
>
> *Lack of Interactivity*: Users must provide the entire layout upfront. This requires the users to visualize the entire scene in advance and estimate the positions and sizes of the layout boxes to generate the desired scene. If users modify any layout elements, the entire generated image is affected, as demonstrated in Figure 2, making the iterative refinement of the image cumbersome.
>
> *Absence of 3D Capabilities*: These approaches do not support placing and rotating objects or the camera in 3D space.
>
> To address these limitations, our approach tackles both issues simultaneously. We enable users to interact with the layout by controlling one object at a time in 3D and refining iteratively before advancing to the next stage. This step-by-step process closely aligns with the natural workflow of artists and helps them reach the desired outcome quickly.

---

> > ### Comment · Reviewer_9MBn · 2024-11-28
> > **content preserving problem**
> >
> > Thank you to the authors for providing additional results and details.
> >
> > However, the newly added 3D control example in Figure 6 exhibits significant content preservation issues, such as inconsistencies in the background and directional changes (e.g., from front to tail). Addressing these issues is crucial to strengthen the framework's overall reliability and applicability.

---

> > > ### Author Response · Authors · 2024-11-28
> > >
> > > Thank you for your response!
> > >
> > > There seems to be a missing context in Figure 6. Images in the figure are generated independently based on different 3D scenes and camera setups to demontrate 3D camera and object control.
> > > Therefore, they were not meant to be consistent, but we just used the same prompts to generate them.
> > > We apologize for the confusion and we will make sure to clarify this context in the revised version of the paper.

---

### Comment · Area_Chair_mVU1 · 2024-11-25
**Please read the rebuttal and reply**

Dear Reviewers,

Thanks again for serving for ICLR, the discussion period between authors and reviewers is approaching (November 27 at 11:59pm AoE), please read the rebuttal and ask questions if you have any. Your timely response is important and highly appreciated.

Thanks,

AC

---

### Meta-Review · Area_Chair_mVU1 · 2024-12-18

**Metareview:**

This paper proposes a T2I model that supports interactive 3D layout editing. The main idea is to divide the scene synthesis process in multiple stages, with each stage generating one object. To make the newly generated object has novel style while preserving other parts of the scene, the paper develops a dynamic self-attention mechanism that blends keys and values from previous stage with the ones in current stage. Furthermore, the paper proposes 3D translation to incorporate scene scale change.

The paper is well written and provides effective solution to an interesting task.

During rebuttal, reviewers raised questions about the motivation, limitation (e.g., 3D awareness), generalization to free-form surfaces and questions about quantitative metrics. The authors have actively replied to all questions, addressing most of these. After rebuttal, 3 out of 4 reviewers agree to accept the paper, while one reviewer does not respond. AC agrees that this paper proposes an interesting task and simple yet effective solution, recommending the paper for acceptance. Meanwhile, AC also agrees that the paper is limited, since the main benefit of 3D layout control is free-form surfaces, the authors are encouraged to explore more in the future.

**Additional Comments On Reviewer Discussion:**

During rebuttal, reviewers raised questions about
- motivation (Reviewer 9MBn)
- limitation (e.g., 3D awareness) (Reviewer 9MBn)
- novelty (Reviewer oNnn)
- efficiency (Reviewer oNnn)
- generalization to free-form surfaces (Reviewer JbbB)
- questions about quantitative metrics. (Reviewer X2dB)

The authors have actively replied to all questions, addressing most of these.

---

### Decision · Program_Chairs · 2025-01-22

Accept (Poster)